# How to calibrate Gaussian two-factor model using swaption

**Myeongsu Choi, Hyoung-Goo Kang***

Business School, Hanyang University, Seoul, Republic of Korea

* hyoungkang@hanyang.ac.kr

## Abstract

We propose an efficient approximation of the swaption normal volatility to estimate the mean reversion separately from the other volatility parameters in the Gaussian two-factor model. We compare our two-step approach with a one-step method that calibrates all parameters simultaneously. The comparison is based on the data from interest rate market of Korea and the US. The parameter estimates of our proposed two-step method are more stable than those of the one-step method in that the latter is overly sensitive to market changes whereas the former is not. The proposed approach also eliminates many existing problems in the Gaussian two-factor model.

**Data Availability Statement:** All relevant data are available within the paper and its Supporting Information files.

**Funding:** Unfunded studies.

## Introduction

This paper aims to detail and address the calibration and parameter-control issues associated with the Gaussian two-factor model (G2PP) in the affine term structure model (ATSM) class. The G2PP model, characterized by two mean reversions and volatility parameters, is prevalent among market practitioners for derivatives pricing and risk management because it is concise and easy to handle analytically. Despite its popularity, previous studies have not reported calibration problems faced while using this model. Hull and White [1] propose the basic procedure for the calibration of the Hull-White model using a tree method. However, there are still some critical issues that this paper aims to address.

First, it is often argued that the adoption of time-dependent parameters in the G2PP model can lead to overfitting. However, prior literature does not present specific cases to elucidate this issue. Consequently, the boundary conditions of the model remain unclear. For example, under what contexts and to what extent can such overfitting concerns be justified? This also raises the question of how to configure the model parameters, given specific applications such as piecewise constant parameters. Second, how do we calibrate model parameters when they are time-dependent? When configuring a specific parameter, we need to determine the strategy to be used when calibrating the model and whether to regard some parameters as time-dependent or not. Third, one of the most critical issues from the trader's point of view is how to control the model parameters in line with their intuitions. Traders often want to incorporate their market views in the pricing model by adjusting the model parameters. However, the existing literature does not provide a method for incorporating traders' views into the model parameters.

**Competing interests:** NO authors have competing interests.

This study addresses these issues. Our approach presents sufficient control over the parameters while allowing sufficient freedom of fitting. We analyze three methods to calibrate the parameters in simultaneous or two-step approaches. The simultaneous method estimates all parameters at once, whereas the two-step method estimates some parameters first and then the others. Next, we propose a method for the approximation of the swaption normal volatilities. Our approach leads to a very efficient approximation, especially for constant model parameters, and provides additional insight into market and model relationships because of its explicit form. We demonstrate the importance of the mean-reversion parameters when fitting several swaption tenors over the entire swaption matrix. We find that the approximation produces stable calibration results when considering both the mean reversions and volatility parameters as constants.

In summary, this study extends the ideas of Hull and White [1,2]. We investigate whether to impose constraints to some time-dependent parameters and analyze various issues such as parameter stability and the relationship between model parameters and the market price (or volatility) of calibration instruments. Therefore, our study is a complementary and detailed follow-up of the work done by Hull and White [1,2].

We highlight ATSM over the market model started by Brace et al. [3] and Jamshidian [4], which has become the standard to price structured interest rate products. In the market model, the dynamics of observable market interest rates such as the London Interbank Offered Rate (LIBOR) and swap rates, are incorporated directly. The model is attractive to market practitioners because it facilitates the calibration of the market cap and swaption volatility and initial yield curves. However, the class of market models has one critical disadvantage. It can be implemented only through simulation, not through a closed form. This simulation is usually slow because several state variables must evolve. It is a typical problem for path-dependent products (i.e., range accrual products) that require the simulation of many paths to achieve sufficient accuracy. The problem can be even more critical for early exercise products (i.e., callable swap and bermudan swaption) because simulation is inconvenient to perform backward calculations to determine the optimal exercise time [5]. Unlike market models, ATSM attempts to capture bond yields by modeling short-term and unobservable interest rates. In this regard, ATSM has several advantageous properties.

First, analytical calculations are possible, admitting closed forms for the cap/floor and efficient price estimation for European swaption. Second, it is simpler for ATSM to implement Monte Carlo simulation than for the market model to do so. Third, in ATSM, all interest rates (i.e., futures and swap rates with different maturity/payment dates) can be easily calculated from short rates. By contrast, market models often require interpolation and extrapolation for dates that do not match the date of the standard model. Many market practitioners still use ATSM for trading, hedging, and risk management, even after market models have been introduced owing to the computational efficiencies provided by ATSM.

The remainder of this paper is organized as follows. Section 2 reviews the G2PP model. It includes analytic formulas required for our analysis, including an approximation of swaption normal volatility. Section 3 suggests how to calibrate models under various configurations. Section 4 presents the numerical results used to test the calibration methods and discusses their performance. Section 5 concludes the paper.

## Model dynamics and closed forms

To evaluate the swaption price analytically, we analyze the variance of the zero-coupon bond ratio and the functions that characterize the affine structure of the bond. We use the closed

formulas found in previous studies [6]. In the G2PP model, the short rate is given by:

$$r(t) = x(t) + y(t) + \theta(t), \tag{1}$$

where $\theta$ is the deterministic function, and $x$, $y$ are stochastic processes. In S1 Appendix, we specify the mean and variance of $x$ and $y$, the zero-coupon bond price, and the swaption price more explicitly.

## Dynamics

Under a risk-neutral measure $Q$, the stochastic differential equations (SDE) of the G2PP model can be written as:

$$
\begin{aligned}
r(t) &= x(t) + y(t) + \theta(t), \ r(0) = r_0, \\
dx(t) &= -a(t)x(t)dt + \sigma(t)dW_1(t), \ x(0) = 0, \\
dy(t) &= -b(t)y(t)dt + \eta(t)dW_2(t), \ \ y(0) = 0,
\end{aligned}
\tag{2}
$$

with the time-dependent functions as follows: $\{a(t), b(t)\}$ for the mean reversions, $\{\sigma(t), \eta(t)\}$ for the volatility parameters, and $<dW_1(t), dW_2(t)> = \rho(t)dt$. The function $\theta(t)$ reflects the exact initial curve. $r(t)$ follows a normal distribution with mean and variance as:

$$
\begin{aligned}
E[r(t)|F_s] &= E[x(t)|F_s] + E[y(t)|F_s] + \theta(t), \\
Var[r(t)|F_s] &= Var[x(t)|F_s] + Var[y(t)|F_s] + 2Cov[x(t), y(t)|F_s],
\end{aligned}
\tag{3}
$$

where $F_s$ is a $\sigma$-field generated by $r(t)$ up to $s$. The zero-coupon bond $P(t, T)$ can be written as a G2PP

$$P(t, T) = \mathcal{A}(t, T)exp\{-\mathcal{B}(a, t, T)x(t) - \mathcal{B}(b, t, T)y(t)\}, \tag{4}$$

where the functions $\mathcal{A}(t, T)$ and $\mathcal{B}(\cdot, t, T)$ are described and explained in Appendix A. Using Ito's lemma,

$$\frac{dP(t, T)}{P(t, T)} = r(t)dt - \sigma(t)\mathcal{B}(a, t, T)dW_1(t) - \eta(t)\mathcal{B}(b, t, T)dW_2(t). \tag{5}$$

To derive the closed forms, we consider the bond ratio with fixing at time $T_F$ and payment at $T_P$ ($t \leq T_F \leq T_P$), which has the dynamics in the $T_P$-forward measure as:

$$d\left(\frac{P(t, T_F)}{P(t, T_P)}\right) = \frac{P(t, T_F)}{P(t, T_P)}\left[\sigma(t)(\mathcal{B}(a, t, T_P) - \mathcal{B}(a, t, T_F))dW_1^{T_P}(t) + \eta(t)(\mathcal{B}(b, t, T_P) - \mathcal{B}(b, t, T_F))dW_2^{T_P}(t)\right]. \tag{6}$$

The integrated variance of $\frac{P(t,T_F)}{P(t,T_P)}$ is

$$
\begin{aligned}
V_p(t, T_F, T_P) &= \int_t^{T_F} \sigma^2(u)(\mathcal{B}(a, u, T_P) - \mathcal{B}(a, u, T_F))^2 du \\
&+ \int_t^{T_F} \eta^2(u)(\mathcal{B}(b, u, T_P) - \mathcal{B}(b, u, T_F))^2 du \\
&+ 2\int_t^{T_F} \rho(u)\sigma(u)\eta(u)(\mathcal{B}(a, u, T_P) - \mathcal{B}(a, u, T_F))(\mathcal{B}(b, u, T_P) - \mathcal{B}(b, u, T_F))du \\
&= V_x(t, T_F)\mathcal{B}(a, T_F, T_P)^2 + V_y(t, T_F)\mathcal{B}(b, T_F, T_P)^2 \\
&+ 2Cov_{x,y}(t, T_F)\mathcal{B}(a, T_F, T_P)\mathcal{B}(B, T_F, T_P),
\end{aligned}
\tag{7}
$$

where $V_x$ and $V_y$ are the variance of $x$ and $y$ respectively, and $Cov_{x,y}$ is the covariance of $x$ and $y$.

**Closed form for caplet and swaption.** Let us consider a caplet with strike $K$, fixing date $T_F$, and paying date $T_P$. The caplet can be written as a zero-coupon bond put option (*ZBP*). Applying the black formula to the variance of the bond ratio produces:

$$Caplet(K, T_F, T_P) = (1 + K\delta)ZBP\left(T_F, T_P, \frac{1}{1 + K\delta}\right),$$

$$ZBP(T_F, T_P, X) = KP(0, T_F)N(d_1) - P(0, T_P)N(d_2),$$

$$d_1 = \frac{ln\left(\frac{P(0, T_F)X}{P(0, T_P)}\right)}{\sqrt{V_P(0, T_F, T_P)}} + \frac{1}{2}\sqrt{V_P(0, T_F, T_P)}, d_2 = d_1 - \sqrt{V_P(0, T_F, T_P)},$$

(8)

where $\delta = \delta(T_F, T_P)$ is the fraction in years from $T_F$ to $T_P$.

Next, let us consider a payer swaption with notional $N$, strike $K$, option expiry $T_0$, swap maturity $T_P$, and swap cash flows at time $\{T_i\}_{i = 1, \cdots, n}$ ($T_P = T_n$). In the G2PP model, Jamshidian's decomposition for the coupon-bearing-bond option and European swaptions are not applicable. Therefore, such products need to be priced via alternative methods, such as numerical integration or Monte Carlo simulations. The swaption price when the model parameters of the G2PP are constant is as follows:

$$PSwaption(K, T_0, T_P)$$

$$= NP(0, T_0)\int_{-\infty}^{\infty} \frac{e^{-\frac{1}{2}\left(\frac{x - \tilde{\mu}_1}{\tilde{\sigma}_1}\right)^2}}{\tilde{\sigma}_1\sqrt{2\Pi}}\left[\Phi(-h_1(x)) - \sum_{i=1}^{n}\lambda_i(x)e^{\kappa_i(x)}\Phi(-h_2(x))\right]dx,$$

(9)

where

$$h_1(x) = \frac{\bar{x} - \tilde{\mu}_2}{\tilde{\sigma}_2\sqrt{1 - \tilde{\rho}^2}} - \frac{\tilde{\rho}(x - \tilde{\mu}_1)}{\tilde{\sigma}_1\sqrt{1 - \tilde{\rho}^2}}, \quad h_2 = h_1(x) + \mathcal{B}(b, T_0, T_i)\tilde{\sigma}_2\sqrt{1 - \tilde{\rho}^2},$$

$$\lambda_i(x) = c_i\mathcal{A}(T_0, T_i)e^{-x\mathcal{B}(a, T_0, T_i)}, \quad \sum_{i=1}^{n}\lambda_i e^{-\bar{x}\mathcal{B}(b, T_0, T_i)} = 1,$$

$$c_i = K\tau_i \text{ for } \alpha < i < \beta \text{ and } c_\beta = 1 + K\tau_\beta,$$

$$\kappa_i(x) = -\mathcal{B}(b, T_0, T_i)\left(\tilde{\mu}_2 - \frac{\tilde{\sigma}_2^2(1 - \tilde{\rho}^2)}{2}\mathcal{B}(b, T_0, T_i) + \tilde{\rho}\tilde{\sigma}_2\frac{x - \mu_1}{\tilde{\sigma}_1}\right),$$

$$\tilde{\mu}_1 = \frac{\sigma^2}{2a^2}(1 - e^{-2aT_0}) + \frac{\sigma\eta\rho}{b}\mathcal{B}(a + b, 0, T_0) - \left(\frac{\sigma^2}{a} + \frac{\sigma\eta\rho}{b}\right)\mathcal{B}(a, 0, T_0),$$

$$\tilde{\mu}_2 = \frac{\eta^2}{2b^2}(1 - e^{-2bT_0}) + \frac{\sigma\eta\rho}{a}\mathcal{B}(a + b, 0, T_0) - \left(\frac{\eta^2}{b} + \frac{\sigma\eta\rho}{a}\right)\mathcal{B}(b, 0, T_0),$$

$$\tilde{\sigma}_1 = \sigma\sqrt{\frac{1 - e^{-2aT_0}}{2a}}, \quad \tilde{\sigma}_2 = \eta\sqrt{\frac{1 - e^{-2bT_0}}{2b}}, \quad \tilde{\rho} = \frac{\sigma\eta\rho}{\tilde{\sigma}_1\tilde{\sigma}_2}\mathcal{B}(a + b, 0, T_0),$$

$$\mathcal{A}(\mathcal{T}_0, \mathcal{T}\ ) = exp\left\{\frac{1}{2}V_P^2(T_0, T_i) - \int_{T_0}^{T_i} \Phi(u)du\right\},$$

$\Phi$: cumulative distribution function of the standard normal distribution.

**An approximation of swaption normal volatility.** The aforementioned equation analytically expresses the swaption price but depends on the implied volatility obtained from the swaption price calculated using the parameters $a(t)$, $b(t)$, $\sigma(t)$, $\eta(t)$, and $\rho(t)$. Hence, to calibrate the model parameters, the swaption prices must be calculated using numerical integration. Conversely, if we can use the normal volatility of swaption in the calibration procedure, there is no need to perform numerical integration. Therefore, this section proposes an approximation to calculate not swaption prices but swaption normal volatilities.

We approximate the swaption normal volatility using the swap market model (SMM). In this model, the swap rate has martingale dynamics. Given the effective date $T_0$ and maturity date $T_n$ with $t<T_0<T_n$, the swaption payoff is

$$V_{swaption}(T_0) = A(T_0)(S(T_0) - K)^+, \tag{10}$$

where $A(t)$ and $S(t)$ are the swap annuity and forward swap rate, respectively:

$$A(t) \triangleq A_{0,N}(t) = \sum_{i=0}^{N-1} \tau_i P(t, T_{i+1}),$$

$$S(t) \triangleq S_{0,N}(t) = \frac{P(t, T_0) - P(t, T_N)}{A(t)}. \tag{11}$$

In the G2PP model, $A(t)$ and $S(t)$ are determined by $x(t)$ and $y(t)$:

$$A(t) = A(t, x(t), y(t)),$$
$$S(t) = S(t, x(t), y(t)). \tag{12}$$

Thus, using Ito's lemma and changing measure from the risk-neutral measure to the annuity measure, we have:

$$dS(t) = drift + \frac{\partial S}{\partial x}(t, x(t), y(t))\sigma(t)dW_1(t) + \frac{\partial S}{\partial y}(t, x(t), y(t))\eta(t)dW_2(t)$$

$$= \sqrt{\left[\frac{\partial S}{\partial x}(\cdot)\sigma(t)\right]^2 + \left[\frac{\partial S}{\partial y}(\cdot)\eta(t)\right]^2 + 2\rho(t)\sigma(t)\eta(t)\frac{\partial S}{\partial x}(\cdot)\frac{\partial S}{\partial y}(\cdot)}dW^A(t) \tag{13}$$

$$= \sigma_S(t)dW^A(t).$$

Because the functions $\frac{\partial S}{\partial x}(t, x(t), y(t)), \frac{\partial S}{\partial y}(t, x(t), y(t))$ are close to constants, we can approximate,

$$\frac{\partial S}{\partial x}(t, x(t), y(t)) \approx \frac{\partial S}{\partial x}(t, \bar{x}(t), \bar{y}(t)),$$

$$\frac{\partial S}{\partial y}(t, x(t), y(t)) \approx \frac{\partial S}{\partial y}(t, \bar{x}(t), \bar{y}(t)), \tag{14}$$

where $(\bar{x}(t), \bar{y}(t))$ is a deterministic proxy of a random vector $(x(t), y(t))$, for instance (0,0). If $\sigma(t)$ and $\eta(t)$ are not too high, then $(\bar{x}(t), \bar{y}(t))$ as (0,0) is a simple choice. Furthermore, it allow replacing $P(t, T, \bar{x}(t), \bar{y}(t))$ with $P(0, T)/P(0, t)$ [6]. Using this formula, we can obtain the

swaption normal volatility directly as:

$$\text{Swaption normal volatility} = \sqrt{\frac{\int_0^{T_0} \sigma_S(t)dt}{T_0}}. \tag{15}$$

## Parameter specifications

To integrate the functions $a(t)$, $b(t)$, $\sigma(t)$, and $\eta(t)$ analytically and easily, we set them to piecewise constant functions. Let us consider the time grid $\{t_i\}_{i=1,\cdots,N-1}$, where $N$ is the number of grid points. Because $a$, $b$, $\sigma$, $\eta$ are piecewise constant, we only need to define the values $\{a_i\}$, $\{b_i\}$, $\{\sigma_i\}$, $\{\eta_i\}$ at $\{t_i\}_{i=1,\cdots,N-1}$ at $[0,+\infty)$.

For example,

$$a(t) = \begin{cases} a_i & t \in [t_i, t_{i+1}) \quad i = 0\cdots, N-1 \\ a_N & t \geq t_N \end{cases}. \tag{16}$$

This can be explained by assuming a time grid running for up to 20 years with 1-year intervals. There are N = 21 grid points, for example, ($t_0 = 0$, $t_{20} = 20$). Therefore, a typical piecewise constant function is defined with 21 values, which means 84 parameters, including mean reversions and volatility parameters. The excessive number of parameters makes the calculations computationally expensive; hence, we need to impose some constraints.

One way to reduce the number of parameters is to match the number of parameters with market instruments using methods such as bootstrap. This strategy is typically used for Hull–White one-factor (HW1F) model; however, it is not suitable for G2PP. In the HW1F model, one parameter exists per basket [7]; however, in G2PP, two or more parameters exist in one calibration basket.

To reduce the number of parameters, we predetermine the correlation coefficient. Andersen and Piterbarg [6] suggest that the forward rate correlation, $\rho(t)$, is not necessarily a monotonic function. For calibration purposes, it is often useful to consider $\rho(t, t, \infty)$, the correlation between the short rate and the long-dated forward rate. Assuming a time-homogeneous correlation structure, we obtain

$$\rho(t, t, \infty) = \frac{1 + \rho\mathcal{C}}{\sqrt{1 + 2\rho\mathcal{C} + \mathcal{C}^2}}, \quad \mathcal{C} = \frac{\eta(t)}{\sigma(t)}, \tag{17}$$

an expression that does not depend on the mean reversions of $x$ and $y$. Assuming that $\mathcal{C}$ is a constant, we can obtain the correlation coefficient $\rho$ from this relationship with no other model parameters. The cms spread option market provides us with information on the correlation between the interest rates. However, in the absence of such a market, this scheme may be useful.

## Calibration methodologies

This section suggests methods to calibrate the G2PP model at the swaption market (volatilities) using the swaption approximation equation presented in Section 2. By calibration methodologies, we mean the following: (1) the choice of constant or time-varying mean reversions and volatilities, (2) the choice of products for calibration, (3) whether to calibrate locally or globally, (4) whether to calibrate the mean reversions and the volatilities at once or separately and in the latter case, how to estimate them independent of the other.

### Relationship between market data and model parameters

Before starting the calibration procedures, it is useful to consider what types of the swaption normal volatilities are obtained from the G2PP model. This section aims to study the effect of mean reversions and volatility parameters on the swaption normal volatility.

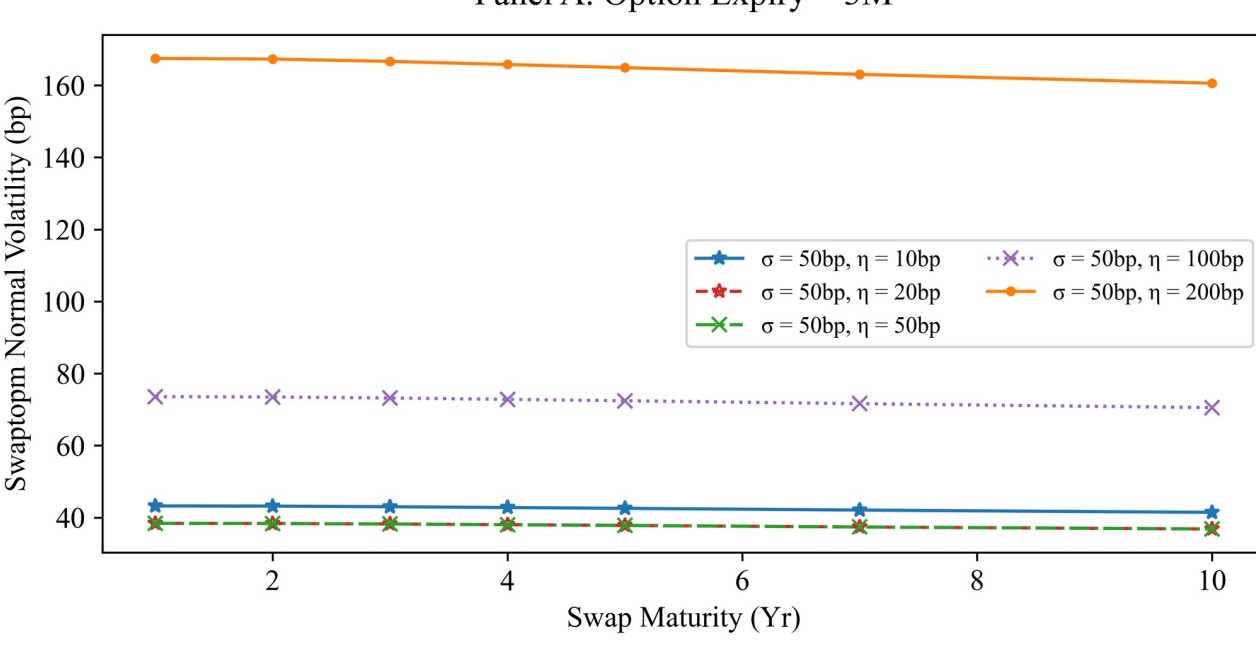

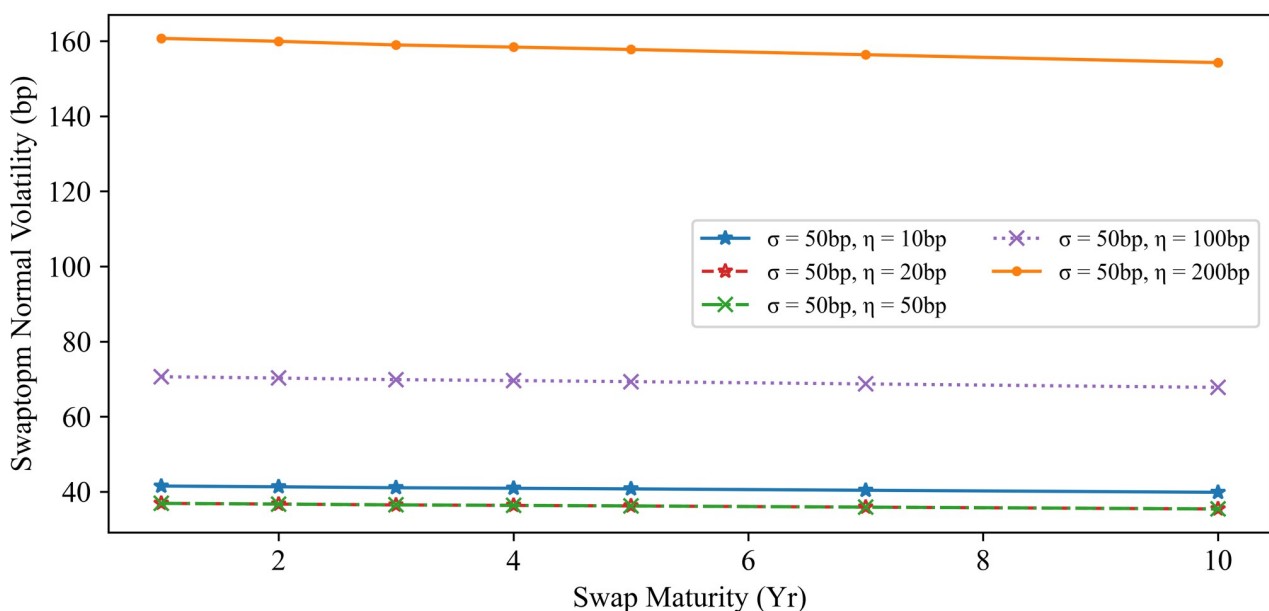

**Fig 1. Swaption normal volatilities for the G2PP model with different model volatilities.** Panel A shows the swaption normal volatilities for the 3-months option expiry in the G2PP model with different volatility parameters and fixed mean reversions (*a*, *b* = 0.01). Panel B shows the swaption normal volatilities for the 10-years option expiry in the G2PP model under the same conditions as those in Panel A. Here, it is difficult to determine the relationship between mean reversions and swaption normal volatility.

We change the value of the constant volatility parameters $\sigma$ and $\eta$ with fixed mean reversions at 0.01. We display the result in Fig 1 for the option expiries of 3M and 10Y. In panel A of Fig 1, it is clear that as the volatility parameters of the model increase, the swaption normal volatility increases. Furthermore, at fixed option expiry, the ratios between two normal

volatilities with swap tenors vary little with volatility parameters. That is, the normal volatility ratio of swaption is independent of the volatility parameters. The volatility parameters σ and η are affected by the level of the normal volatility curves, without significantly changing their shapes. More details are given in next Section 4.1.

Next, we fix the volatilities parameters and change the mean reversions to constants. The results are shown in Fig 2, which indicates that 0 have a qualitatively different effect on the swaption normal volatility. When the change between 0.5% and 20% is considered, we see that the swaption normal volatility curve decreases monotonically as the swap maturity increases, unless the mean reversion is 20%.

## Calibration on the mean reversion

Market practitioners tend to implement two methodologies to calibrate the HW1F model for vanilla products such as callable swap. The first method calibrates both the mean reversions and the volatility parameters simultaneously. The second calibrates the mean reversions first and then calibrates the volatility parameters for cap/floor and/or swaption volatilities.

We compare two strategies to calibrate the G2PP model. The first calibrates all parameters, including ρ simultaneously. The second one that we propose fixes ρ, $a$, $b$ and then calibrates only the volatility parameters to swaption. We present detailed numerical examples of the second method. Note that this study does not aim to identify which method is better. Rather, we present the strategies and collect some numerical results. We believe our results are useful to practitioners, and to the best of our knowledge, are the first to be reported in the literature.

## Bootstrap vs. global calibration

The bootstrap method is the first strategy we consider. This method uses the first market instrument to estimate model parameters from 0 to the first maturity $T_1$, and the second market instrument to estimate model parameters from $T_1$ to $T_2$, etc.

In the first-time grid, up to five parameters exist that need to be calibrated. However, this often results in a large change in parameters. Moreover, the optimization results highly depend on the initial seeds owing to the curse of dimensionality. To avoid such disadvantages, we suggest the implementation of the following strategy.

First, we do not perfectly fit specific instruments for calibration; rather, we consider them as baskets for optimization. Second, we choose parametric forms for the mean reversions and volatility parameters excluding the correlation coefficient ρ, as explained in the previous section. The number of model parameters and fitted instruments are not necessarily equal. Next, we calibrate all model parameters (except $\rho$) to fit all the market instruments simultaneously, between the model and market volatilities.

We can add constraints to the model parameters in the parametric forms for the calibration basket. Therefore, if we take the functionals suitably, such as the addition of a penalty function with boundary conditions, no big jumps will appear. However, a disadvantage is that it is difficult to find how one parameter affect the parametric forms for a calibration basket. Generally, we must perform multi-dimensional optimizations, such as the Levenberg–Marquardt algorithm [8]. A goal of this study is to show that imposing constraints on model parameters produces more stable and meaningful results.

## Calibration to swaptions

This section introduces three calibration methods. The key difference among the methods is the way the mean reversions are handled regardless of whether the parameters are constant or a function of time. First, we estimate the mean reversions independently of the volatility

## Panel A. Option Expiry = 3M

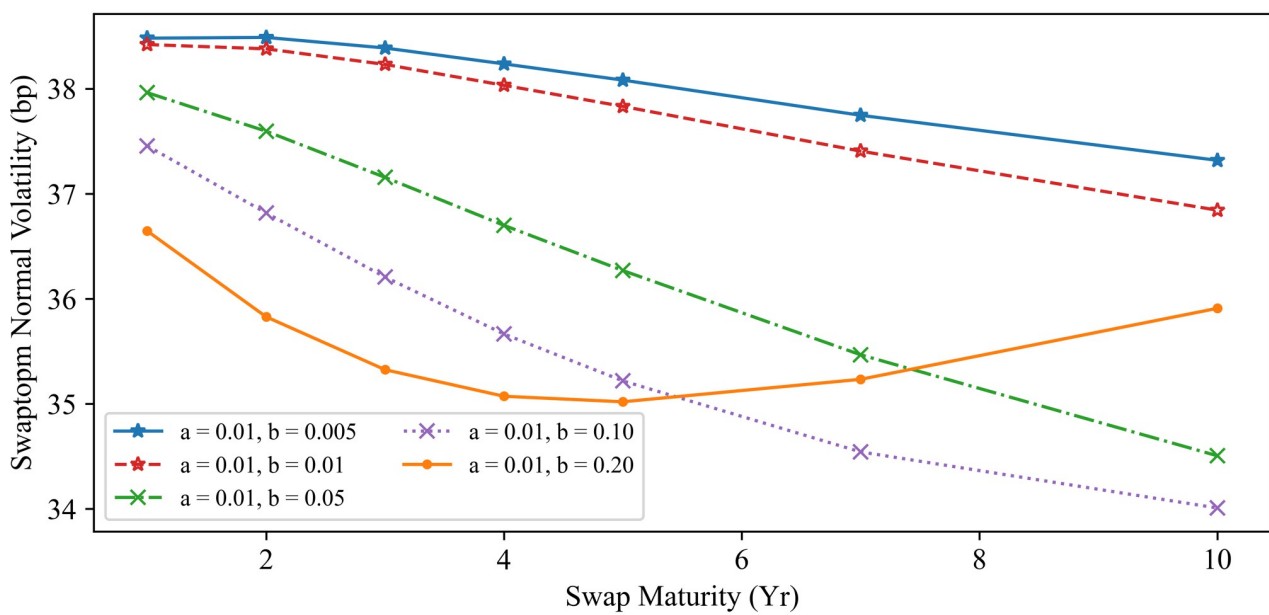

## Panel B. Option Expiry = 10Y

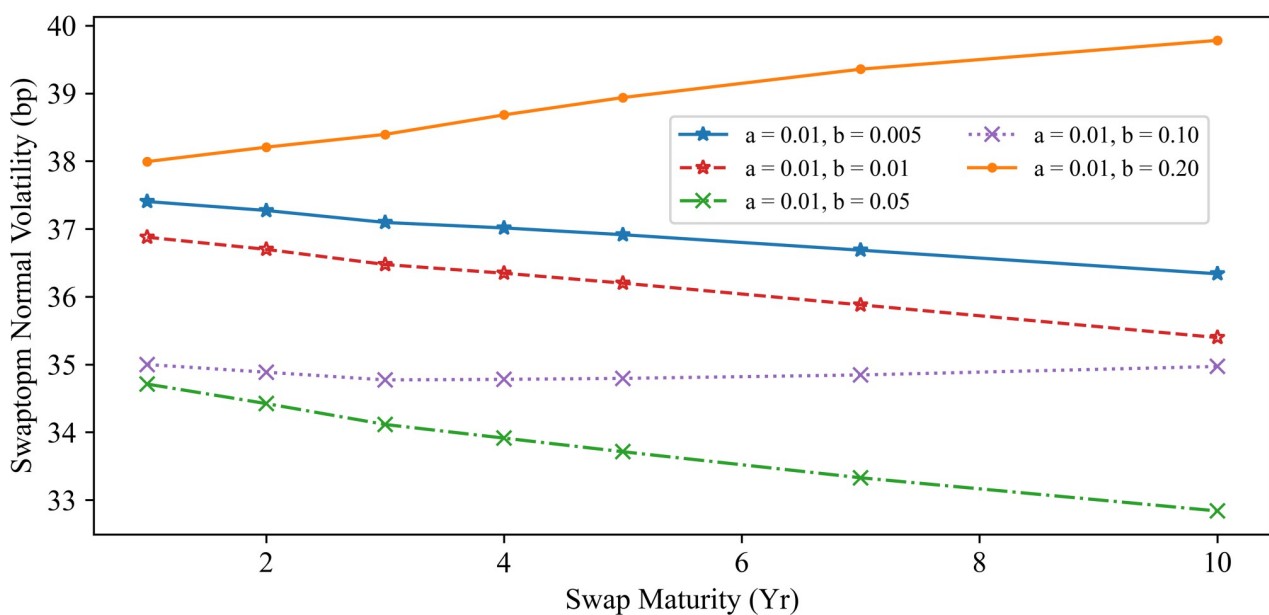

**Fig 2. Swaption normal volatilities for the G2PP model with different mean reversions.** Panel A shows the swaption normal volatilities for the 3-months option expiry in the G2PP model with different mean reversions and fixed volatility parameters ($\sigma = 50bp$, $\eta = 50bp$). Panel B shows the swaption normal volatilities for the 10-years option expiry in the G2PP model under the same conditions as those in Panel A. Here, it is difficult to determine the relationship between mean reversions and swaption normal volatility.

parameters and then find the volatility parameters. The second method simultaneously estimates the mean reversions and volatility parameters with a fixed correlation coefficient. In the third method, the mean reversions, volatility parameters, and correlation coefficient are all estimated simultaneously. All optimizations to calibrate the parameters are performed using

the Levenberg–Marquardt algorithm. The user must select the initial seed of the optimization. In practice, the latest calibration results are used as the initial seed. These methods are described in the following subsections.

**Method I (Two-step approach).** The first method (Method I hereafter) comprises two steps: (1) Calibrate the mean reversion parameters $\{a, b\}$ using the volatility ratios of swaptions described further in this section. (2) Calibrate on the volatility using the SMM approximation with the mean reversions in (1).

In step (1), by making two assumptions, we do not need the volatility parameters of G2PP. We can only consider the ratio of two swap variances with the same option expiry $M_i$ but different swap tenors $T_j$ and $T_k$. Thus,

$$
\frac{Var(M_i, T_j)}{Var(M_i, T_k)} \approx \frac{\left( \begin{array}{c} \mathcal{C}^2 C_a^2(M_i, T_j) \int_0^{M_i} e^{2at}dt + C_b^2(M_i, T_j) \int_0^{M_i} e^{2bt}dt \\ +2\mathcal{C}C_a(M_i, T_j)C_b(M_i, T_j) \int_0^{M_i} e^{2(a+b)t}dt \end{array} \right)}{\left( \begin{array}{c} \mathcal{C}^2 C_a^2(M_i, T_k) \int_0^{M_i} e^{2at}dt + C_b^2(M_i, T_k) \int_0^{M_i} e^{2bt}dt \\ +2\mathcal{C}C_a(M_i, T_k)C_b(M_i, T_k) \int_0^{M_k} e^{2(a+b)t}dt \end{array} \right)}, \tag{18}
$$

where

$$
C_a(M_i, T_k) = \frac{e^{-aM_i}P(0, M_i) - e^{-aT_k}P(0, T_k) - S(0)\sum_{i=0}^{N-1} \tau_i P(0, T_{i+1})e^{-aT_{i+1}}}{aA(0)},
$$

$$
\mathcal{C} = \frac{\sigma}{\eta},
$$

which is independent of the volatility parameters. Using the volatility ratio of swaption between model and market, we can obtain the mean reversions without the volatility parameters $\{\sigma, \eta\}$ of G2PP. More precisely, we can use the market swaption normal volatilities with option expiry $M_i$ and swap tenor $T_j$ such that there are $n_m$ option expiries and $n_t$ swap tenors. We can calibrate the mean reversions $\{a, b\}$ (possibly multi-dimensional) at which the function

$$
G(a, b) = \sum_{\substack{1 \le i \le n_m, \\ 1 \le j, k \le n_{t-1}}} \left( \sqrt{\frac{Var(M_i, T_j)}{Var(M_i, T_k)}} - \frac{\mathcal{V}^{mkt}(M_i, T_j)}{\mathcal{V}^{mkt}(M_i, T_k)} \right)^2, \tag{19}
$$

has its minimum value.

Step (2) is an estimation of the volatility parameters. We only estimate the volatility parameters, such that the analytical swaption normal volatility $\mathcal{V}_{a,b,\rho}^{mod}(M_i, T_j)(\sigma, \eta)$ matches the swaption normal volatility $\mathcal{V}^{mkt}(M_i, T_j)$ from the market. $\mathcal{V}^{mod}$ is an approximation formula that calculates the swaption normal volatility in Section 2.3 using at-the-money (ATM) options. The subscripts for $\{a, b, \rho\}$ indicate that the mean reversions and correlation coefficient have been fixed at predefined values. That is, our volatilities are the points $\sigma, \eta$ at which the function

$$
G_{a,b,\rho}(\sigma, \eta) = \sum_{\substack{1 \le i \le n_m \\ 1 \le j \le n_{t-1}}} (\mathcal{V}_{a,b,\rho}^{mod}(M_i, T_j)(\sigma, \eta) - \mathcal{V}^{mkt}(M_i, T_j))^2, \tag{20}
$$

has a minimum value. This method requires the performance of two two-dimensional (2-D) optimizations.

**Method II.** The second method (Method II) is very simple with just one step: calibration of the mean reversions and the volatility parameters simultaneously. The calibrated mean reversions and volatility parameters are the points at which function

$$G_{\rho}(a, b, \sigma, \eta) = \sum_{\substack{1 \leq i \leq n_m \\ 1 \leq j \leq n_{t-1}}} (\mathcal{V}_{\rho}^{mod}(M_i, T_j)(a, b, \sigma, \eta) - \mathcal{V}^{mkt}(M_i, T_j))^2, \tag{21}$$

has its minimum value. This method means that we perform one four-dimensional (4-D) minimization. It is more sensitive to the initial seed than Method I because there are more targets to optimize.

**Method III.** In the third method (Method III), the mean reversions, volatility parameters, and correlation coefficient are calibrated simultaneously. This method involves performing one five-dimensional (5-D) minimization with the following objective function:

$$G(a, b, \rho, \sigma, \eta) = \sum_{\substack{1 \leq i \leq n_m \\ 1 \leq j \leq n_{t-1}}} (\mathcal{V}^{mod}(M_i, T_j) - \mathcal{V}^{mkt}(M_i, T_j))^2. \tag{22}$$

It is more sensitive to the initial seed than Method I and Method II because it has more targets to optimize.

## Results

This section shows the numerical results. We apply the methods described in Section 3 to the Korea Won ATM swaption. The swaptions dataset comprises the swaption normal volatility of option expiries 1-, 3-, 6-, 9-, 18-months, 1-, 2-, 3-, 4-, 5-, 7-, and 10-years, with underlying swap maturities of 1-, 2-, 3-, 4-, 5-, 7-, and 10-years each (84 swaption contracts exist in total). To follow the market convention, a swaption is considered ATM option when the strike rate equals the forward swap rate for the same swap maturity. To construct the yield curve, we use 1-day call rate, 3-months Certificates of Deposit (CD) rate, and swap rate with maturity 6-, 9-, 18-months, 1-, 2-, 3-, 4-, 5-, 7-, 10-, 11-, 12-, 15-, 20-, 25-, 30-years.

The summary statistics of the KRW yield curve from January 2017 to December 2020 are reported in Panel A of Table 1. The number of observations is 997. Panel B, C, D, and E of Table 1 show the summary statistics of KRW swaption normal volatility for the same period. We obtain all data from the Refinitiv composite page quoted by the broker [9].

### Swaption normal volatility by SMM approximation

The swaption normal volatility using the SMM approximation in Section 2.3 shows the effect of the mean reversions and volatility parameters. Using this approximation, we can determine the relationship between each model parameter and the swaption normal volatilities.

Fig 1 shows the swaption normal volatilities using the SMM approximation corresponding to different volatility parameter $\eta$ values (10, 20, 50, 100, and 200 bp) while fixing the mean reversions at 0.01, and the other volatility parameter $\sigma$ at 50 bp. This figure shows that the swaption normal volatilities with high $\eta$ are larger than those with low volatility parameters.

Fig 2 shows the swaption normal volatilities calculated using SMM approximations according to different mean reversion $b$ values (0.005, 0.01, 0.05, 0.10, and 0.20) while fixing the volatility parameters at 50 bp and the other mean reversion $a$ at 0.01. The relationship between the mean reversion and swaption normal volatility is inconclusive.

Panel A of Figs 3 and 4 display the ratios of the swaption normal volatilities with fixed $\mathcal{C} = \sigma/\eta$ under the mean reversion parameters. Panel B of Figs 3 and 4 focuses on the influence of changing the volatilities on the normal volatility ratios with fixed mean reversions. All

**Table 1. Summary of statistics.** Panel A reports the summary statistics of the KRW yield curve from January 2017 to December 2020. The number of observations is 997. Panels B, C, D, and E show the summary statistics of KRW swaption normal volatility for the same period.

### Panel A. KRW Yield Curve (%)

| | O/N | 3M | 6M | 9M | 1Y | 18M | 2Y | 3Y | 4Y | 5Y | 6Y | 7Y | 8Y | 9Y | 10Y | 11Y | 12Y | 15Y | 20Y | 25Y | 30Y |
|---|---|---|---|---|---|---|---|---|---|---|---|---|---|---|---|---|---|---|---|---|---|
| count | 997 | 997 | 997 | 997 | 997 | 997 | 997 | 997 | 997 | 997 | 997 | 997 | 997 | 997 | 997 | 997 | 997 | 997 | 997 | 997 | 997 |
| mean | 1.2566 | 1.4246 | 1.4327 | 1.4433 | 1.4538 | 1.4784 | 1.4965 | 1.5338 | 1.5671 | 1.5922 | 1.6088 | 1.6244 | 1.6430 | 1.6632 | 1.6827 | 1.6966 | 1.7091 | 1.7017 | 1.6624 | 1.6498 | 1.6471 |
| std | 0.4040 | 0.3695 | 0.3770 | 0.3882 | 0.4005 | 0.4153 | 0.4273 | 0.4401 | 0.4465 | 0.4537 | 0.4600 | 0.4646 | 0.4700 | 0.4729 | 0.4749 | 0.4812 | 0.4846 | 0.5075 | 0.5561 | 0.5710 | 0.5754 |
| min | 0.4300 | 0.6300 | 0.6525 | 0.6700 | 0.6800 | 0.7000 | 0.7225 | 0.7600 | 0.7825 | 0.8025 | 0.8125 | 0.8200 | 0.8225 | 0.8400 | 0.8550 | 0.8625 | 0.8650 | 0.8350 | 0.7025 | 0.6625 | 0.6525 |
| 25% | 1.2300 | 1.3800 | 1.3800 | 1.3275 | 1.2925 | 1.2500 | 1.1950 | 1.1600 | 1.1500 | 1.1575 | 1.1525 | 1.1725 | 1.1925 | 1.2125 | 1.2300 | 1.2425 | 1.2575 | 1.2350 | 1.1725 | 1.1450 | 1.1350 |
| 50% | 1.2700 | 1.4900 | 1.4800 | 1.4825 | 1.5075 | 1.5425 | 1.5675 | 1.6250 | 1.6650 | 1.7000 | 1.7275 | 1.7475 | 1.7700 | 1.7900 | 1.8100 | 1.8250 | 1.8375 | 1.8425 | 1.8175 | 1.8175 | 1.8175 |
| 75% | 1.5100 | 1.6500 | 1.7250 | 1.7775 | 1.8200 | 1.8550 | 1.8750 | 1.9075 | 1.9400 | 1.9700 | 1.9950 | 2.0150 | 2.0375 | 2.0575 | 2.0825 | 2.1025 | 2.1150 | 2.1225 | 2.1325 | 2.1325 | 2.1350 |
| max | 1.8900 | 1.9300 | 1.9125 | 1.9125 | 1.9200 | 2.0250 | 2.1150 | 2.2225 | 2.2950 | 2.3400 | 2.3675 | 2.3950 | 2.4300 | 2.4600 | 2.4850 | 2.5050 | 2.5225 | 2.5400 | 2.5600 | 2.5600 | 2.5600 |

### Panel B. KRW Swaption Normal Volatility with Option Expiry 1M, 3M, 6M (%)

| | 1M1Y | 1M2Y | 1M3Y | 1M4Y | 1M5Y | 1M7Y | 1M10Y | 3M1Y | 3M2Y | 3M3Y | 3M4Y | 3M5Y | 3M7Y | 3M10Y | 6M1Y | 6M2Y | 6M3Y | 6M4Y | 6M5Y | 6M7Y | 6M10Y |
|---|---|---|---|---|---|---|---|---|---|---|---|---|---|---|---|---|---|---|---|---|---|
| count | 997 | 997 | 997 | 997 | 997 | 997 | 997 | 997 | 997 | 997 | 997 | 997 | 997 | 997 | 997 | 997 | 997 | 997 | 997 | 997 | 997 |
| mean | 0.3103 | 0.3512 | 0.3917 | 0.4298 | 0.4734 | 0.4896 | 0.5144 | 0.3459 | 0.3771 | 0.4098 | 0.4430 | 0.4756 | 0.4881 | 0.5202 | 0.3524 | 0.3816 | 0.4124 | 0.4415 | 0.4698 | 0.4839 | 0.5132 |
| std | 0.0675 | 0.0593 | 0.0578 | 0.0640 | 0.0739 | 0.0785 | 0.0862 | 0.0616 | 0.0570 | 0.0579 | 0.0616 | 0.0698 | 0.0748 | 0.0815 | 0.0601 | 0.0555 | 0.0531 | 0.0540 | 0.0605 | 0.0658 | 0.0732 |
| min | 0.1739 | 0.2326 | 0.2739 | 0.2925 | 0.3243 | 0.3266 | 0.3356 | 0.2324 | 0.2716 | 0.2920 | 0.3079 | 0.3247 | 0.3270 | 0.3384 | 0.2375 | 0.2709 | 0.2910 | 0.3203 | 0.3268 | 0.3300 | 0.3432 |
| 25% | 0.2737 | 0.3175 | 0.3554 | 0.3924 | 0.4260 | 0.4382 | 0.4580 | 0.3023 | 0.3338 | 0.3695 | 0.4078 | 0.4330 | 0.4436 | 0.4724 | 0.3118 | 0.3430 | 0.3802 | 0.4111 | 0.4392 | 0.4498 | 0.4730 |
| 50% | 0.3119 | 0.3495 | 0.3846 | 0.4220 | 0.4637 | 0.4815 | 0.5058 | 0.3514 | 0.3764 | 0.4030 | 0.4340 | 0.4660 | 0.4766 | 0.5057 | 0.3527 | 0.3822 | 0.4056 | 0.4324 | 0.4615 | 0.4765 | 0.5022 |
| 75% | 0.3508 | 0.3860 | 0.4211 | 0.4610 | 0.5159 | 0.5331 | 0.5675 | 0.3888 | 0.4134 | 0.4417 | 0.4703 | 0.5074 | 0.5246 | 0.5666 | 0.3947 | 0.4196 | 0.4398 | 0.4672 | 0.5018 | 0.5181 | 0.5541 |
| max | 0.5566 | 0.5743 | 0.6001 | 0.7084 | 0.8196 | 0.8523 | 0.9017 | 0.5591 | 0.5751 | 0.6434 | 0.7307 | 0.8197 | 0.8529 | 0.9013 | 0.5585 | 0.5733 | 0.6195 | 0.6938 | 0.7697 | 0.8073 | 0.8651 |

### Panel C. KRW Swaption Normal Volatility with Option Expiry, 9M, 1Y, 18M (%)

| | 9M1Y | 9M2Y | 9M3Y | 9M4Y | 9M5Y | 9M7Y | 9M10Y | 1Y1Y | 1Y2Y | 1Y3Y | 1Y4Y | 1Y5Y | 1Y7Y | 1Y10Y | 18M1Y | 18M2Y | 18M3Y | 18M4Y | 18M5Y | 18M7Y | 18M10Y |
|---|---|---|---|---|---|---|---|---|---|---|---|---|---|---|---|---|---|---|---|---|---|
| count | 997 | 997 | 997 | 997 | 997 | 997 | 997 | 997 | 997 | 997 | 997 | 997 | 997 | 997 | 997 | 997 | 997 | 997 | 997 | 997 | 997 |
| mean | 0.3624 | 0.3879 | 0.4142 | 0.4403 | 0.4664 | 0.4786 | 0.5039 | 0.3728 | 0.3946 | 0.4179 | 0.4409 | 0.4616 | 0.4741 | 0.4954 | 0.3946 | 0.4128 | 0.4272 | 0.4426 | 0.4567 | 0.4660 | 0.4860 |
| std | 0.0563 | 0.0518 | 0.0495 | 0.0507 | 0.0550 | 0.0596 | 0.0656 | 0.0541 | 0.0503 | 0.0476 | 0.0468 | 0.0499 | 0.0535 | 0.0593 | 0.0477 | 0.0452 | 0.0430 | 0.0425 | 0.0446 | 0.0478 | 0.0525 |
| min | 0.2650 | 0.2801 | 0.3025 | 0.3223 | 0.3425 | 0.3465 | 0.3593 | 0.2681 | 0.2910 | 0.3135 | 0.3395 | 0.3580 | 0.3662 | 0.3750 | 0.3108 | 0.3281 | 0.3399 | 0.3533 | 0.3641 | 0.3671 | 0.377 |
| 25% | 0.3143 | 0.3476 | 0.3823 | 0.4117 | 0.4397 | 0.4504 | 0.4715 | 0.3226 | 0.3550 | 0.3868 | 0.4152 | 0.4397 | 0.4498 | 0.4655 | 0.3536 | 0.3784 | 0.3987 | 0.4183 | 0.4355 | 0.4439 | 0.4618 |
| 50% | 0.3581 | 0.3840 | 0.4082 | 0.4321 | 0.4592 | 0.4718 | 0.4990 | 0.3668 | 0.3842 | 0.4094 | 0.4364 | 0.4549 | 0.4677 | 0.4904 | 0.3849 | 0.3993 | 0.4180 | 0.4375 | 0.4522 | 0.4609 | 0.4805 |
| 75% | 0.3977 | 0.4210 | 0.4405 | 0.4628 | 0.4910 | 0.5057 | 0.5336 | 0.4108 | 0.4230 | 0.4392 | 0.4578 | 0.4791 | 0.4944 | 0.5195 | 0.4247 | 0.4398 | 0.4461 | 0.4572 | 0.4734 | 0.4832 | 0.5055 |
| max | 0.5536 | 0.5581 | 0.6099 | 0.6756 | 0.7423 | 0.7766 | 0.8285 | 0.5648 | 0.5606 | 0.6024 | 0.6642 | 0.7249 | 0.7452 | 0.7899 | 0.5474 | 0.5524 | 0.6027 | 0.6552 | 0.7076 | 0.7191 | 0.7653 |

### Panel D. KRW Swaption Normal Volatility with Option Expiry 2Y, 3Y, 4Y (%)

| | 2Y1Y | 2Y2Y | 2Y3Y | 2Y4Y | 2Y5Y | 2Y7Y | 2Y10Y | 3Y1Y | 3Y2Y | 3Y3Y | 3Y4Y | 3Y5Y | 3Y7Y | 3Y10Y | 4Y1Y | 4Y2Y | 4Y3Y | 4Y4Y | 4Y5Y | 4Y7Y | 4Y10Y |
|---|---|---|---|---|---|---|---|---|---|---|---|---|---|---|---|---|---|---|---|---|---|
| count | 997 | 997 | 997 | 997 | 997 | 997 | 997 | 997 | 997 | 997 | 997 | 997 | 997 | 997 | 997 | 997 | 997 | 997 | 997 | 997 | 997 |
| mean | 0.4216 | 0.4312 | 0.4374 | 0.4440 | 0.4515 | 0.4591 | 0.4771 | 0.4421 | 0.4429 | 0.4413 | 0.4428 | 0.4459 | 0.4510 | 0.4632 | 0.4559 | 0.4502 | 0.4446 | 0.4429 | 0.4430 | 0.4470 | 0.4570 |
| std | 0.0436 | 0.0414 | 0.0394 | 0.0387 | 0.0397 | 0.0421 | 0.0462 | 0.0410 | 0.0387 | 0.0362 | 0.0351 | 0.0350 | 0.0357 | 0.0382 | 0.0399 | 0.0358 | 0.0326 | 0.0304 | 0.0286 | 0.0284 | 0.0300 |
| min | 0.3512 | 0.3609 | 0.3637 | 0.3672 | 0.3701 | 0.3715 | 0.3790 | 0.3691 | 0.3647 | 0.3562 | 0.3667 | 0.3713 | 0.3742 | 0.3799 | 0.3827 | 0.3862 | 0.3786 | 0.3772 | 0.3779 | 0.3824 | 0.3872 |
| 25% | 0.3903 | 0.4011 | 0.4103 | 0.4196 | 0.4298 | 0.4416 | 0.4580 | 0.4085 | 0.4125 | 0.4142 | 0.4198 | 0.4275 | 0.4336 | 0.4421 | 0.4232 | 0.4215 | 0.4196 | 0.4216 | 0.4253 | 0.4306 | 0.4380 |
| 50% | 0.4051 | 0.4161 | 0.4277 | 0.4399 | 0.4490 | 0.4544 | 0.4729 | 0.4393 | 0.4407 | 0.4404 | 0.4409 | 0.4445 | 0.4483 | 0.4613 | 0.4612 | 0.4523 | 0.4458 | 0.4421 | 0.4421 | 0.4452 | 0.4575 |
| 75% | 0.4492 | 0.4555 | 0.4563 | 0.4582 | 0.4670 | 0.4733 | 0.4932 | 0.4668 | 0.4640 | 0.4584 | 0.4557 | 0.4576 | 0.4657 | 0.4779 | 0.4822 | 0.4704 | 0.4612 | 0.4575 | 0.4563 | 0.4607 | 0.4703 |
| max | 0.5423 | 0.5660 | 0.6069 | 0.6507 | 0.6946 | 0.7025 | 0.7396 | 0.5504 | 0.5740 | 0.6028 | 0.6353 | 0.6661 | 0.6668 | 0.6912 | 0.5823 | 0.5841 | 0.5876 | 0.5895 | 0.5931 | 0.6088 | 0.6353 |

(Continued)

**Table 1.** (Continued)

|  | Panel E. KRW Swaption Normal Volatility with Option Expiry 5Y, 7Y, 10Y (%) | | | | | | | | | | | | | | | | | | | | |
|---|---|---|---|---|---|---|---|---|---|---|---|---|---|---|---|---|---|---|---|---|---|
|  | 5Y1Y | 5Y2Y | 5Y3Y | 5Y4Y | 5Y5Y | 5Y7Y | 5Y10Y | 7Y1Y | 7Y2Y | 7Y3Y | 7Y4Y | 7Y5Y | 7Y7Y | 7Y10Y | 10Y1Y | 10Y2Y | 10Y3Y | 10Y4Y | 10Y5Y | 10Y7Y | 10Y10Y |
| count | 997 | 997 | 997 | 997 | 997 | 997 | 997 | 997 | 997 | 997 | 997 | 997 | 997 | 997 | 997 | 997 | 997 | 997 | 997 | 997 | 997 |
| mean | 0.4665 | 0.4572 | 0.4489 | 0.4425 | 0.4406 | 0.4438 | 0.4524 | 0.4670 | 0.4577 | 0.4527 | 0.4429 | 0.4359 | 0.4389 | 0.4495 | 0.4663 | 0.4578 | 0.4500 | 0.4435 | 0.4361 | 0.4402 | 0.4503 |
| std | 0.0417 | 0.0371 | 0.0320 | 0.0271 | 0.0243 | 0.0227 | 0.0236 | 0.0413 | 0.0355 | 0.0308 | 0.0276 | 0.0248 | 0.0249 | 0.0263 | 0.0403 | 0.0353 | 0.0322 | 0.0292 | 0.0254 | 0.0272 | 0.0309 |
| min | 0.3926 | 0.3892 | 0.3841 | 0.3830 | 0.3791 | 0.3862 | 0.3956 | 0.3908 | 0.3985 | 0.3938 | 0.3800 | 0.3772 | 0.3866 | 0.3969 | 0.3934 | 0.4004 | 0.3942 | 0.3834 | 0.3794 | 0.3867 | 0.3971 |
| 25% | 0.4314 | 0.4278 | 0.4239 | 0.4224 | 0.4250 | 0.4301 | 0.4371 | 0.4295 | 0.4277 | 0.4271 | 0.4234 | 0.4193 | 0.4210 | 0.4305 | 0.4304 | 0.4271 | 0.4254 | 0.4211 | 0.4177 | 0.4191 | 0.4265 |
| 50% | 0.4725 | 0.4612 | 0.4498 | 0.4409 | 0.4376 | 0.4426 | 0.4528 | 0.4681 | 0.4605 | 0.4524 | 0.4399 | 0.4326 | 0.4370 | 0.4453 | 0.4689 | 0.4562 | 0.4416 | 0.4400 | 0.4366 | 0.4390 | 0.4455 |
| 75% | 0.4924 | 0.4787 | 0.4679 | 0.4580 | 0.4543 | 0.4555 | 0.4640 | 0.4941 | 0.4792 | 0.4713 | 0.4590 | 0.4498 | 0.4518 | 0.4659 | 0.4924 | 0.4810 | 0.4724 | 0.4617 | 0.4493 | 0.4574 | 0.4737 |
| max | 0.6133 | 0.5966 | 0.5799 | 0.5646 | 0.5497 | 0.5621 | 0.5796 | 0.6129 | 0.5958 | 0.5828 | 0.5702 | 0.5561 | 0.5679 | 0.5866 | 0.6130 | 0.6002 | 0.5870 | 0.5769 | 0.5661 | 0.5784 | 0.5973 |

## Panel A. Changing a, b at fixed σ, η

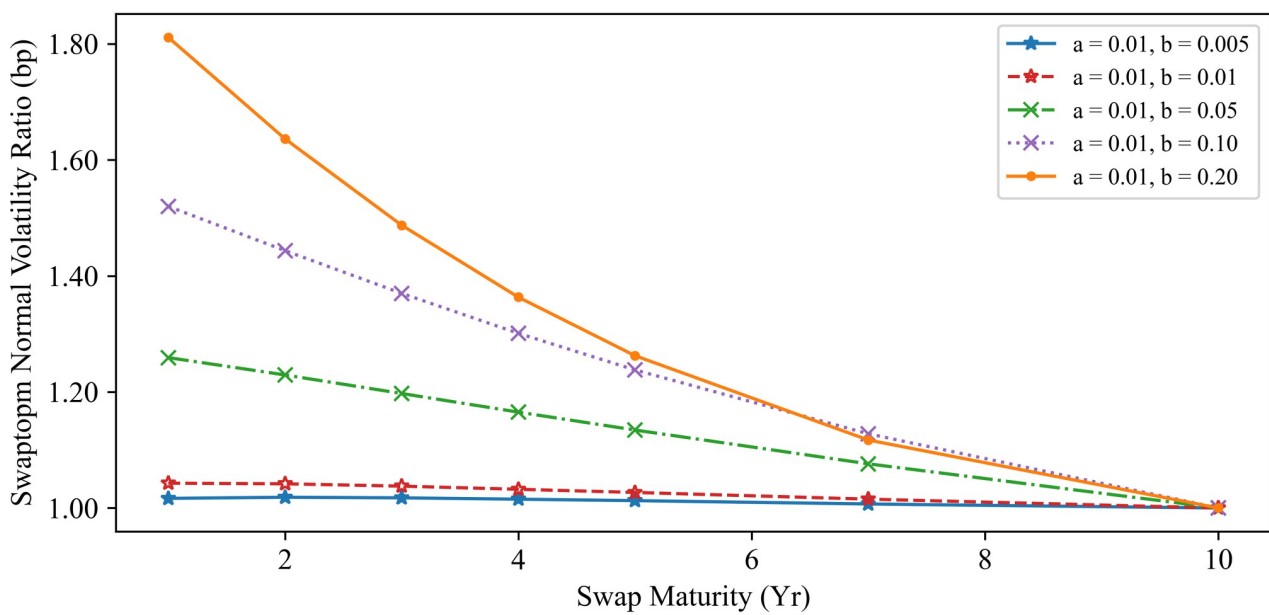

## Panel B. Changing σ, η at fixed a, b

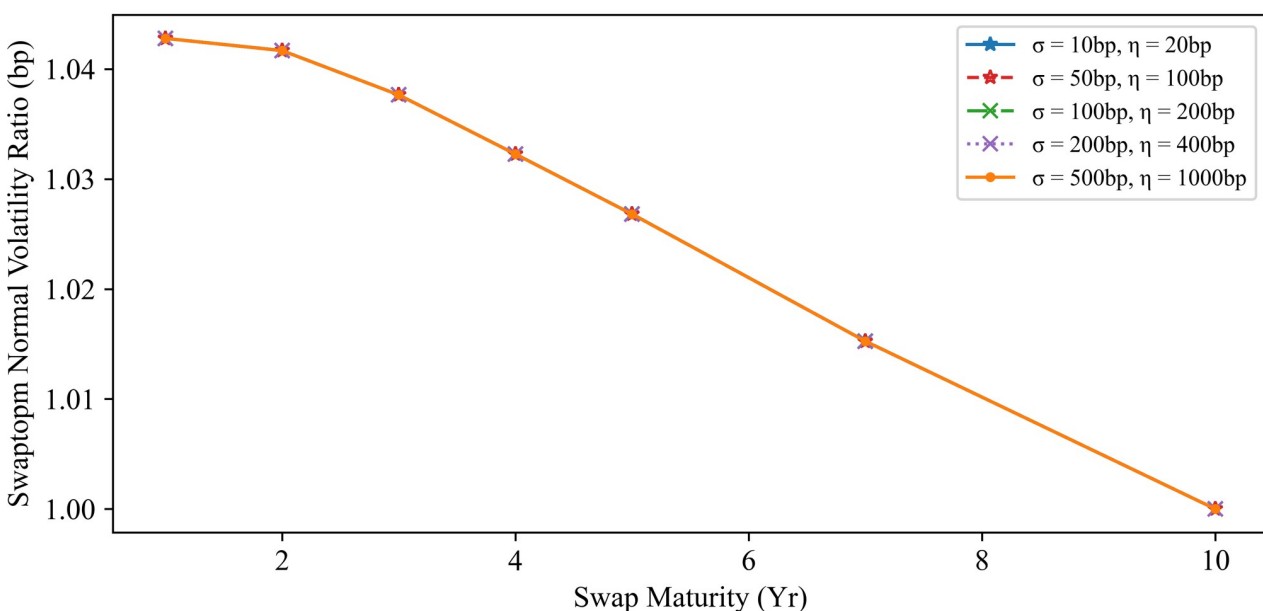

**Fig 3. Swaption normal volatility ratios to swap maturity of 10Y with option expiry of 3M.** Panel A shows the swaption normal volatility ratios to swap maturity 10Y and other tenors under various mean reversions and fixed volatility parameters. All swaptions had an option expiry of 3M. The swaption normal volatility ratio curve moves downward as the mean reversion parameters increase. Panel B shows the swaption normal volatility ratios to swap maturity of 10Y and other tenors under various volatility parameters when the mean reversions and the ratio of volatility parameters are fixed. The swaption normal volatility ratio curves for the various volatility parameters are above the same line. That is, the swaption normal volatility ratios with equal swap maturity depend entirely on the mean reversions.

swaption normal volatilities used in Figs 3 and 4 are computed using the SMM approximation. The swaption normal volatility ratios are insensitive to the volatility parameters changes; they depend mainly on the mean reversions.

## Panel A. Changing a, b at fixed σ, η

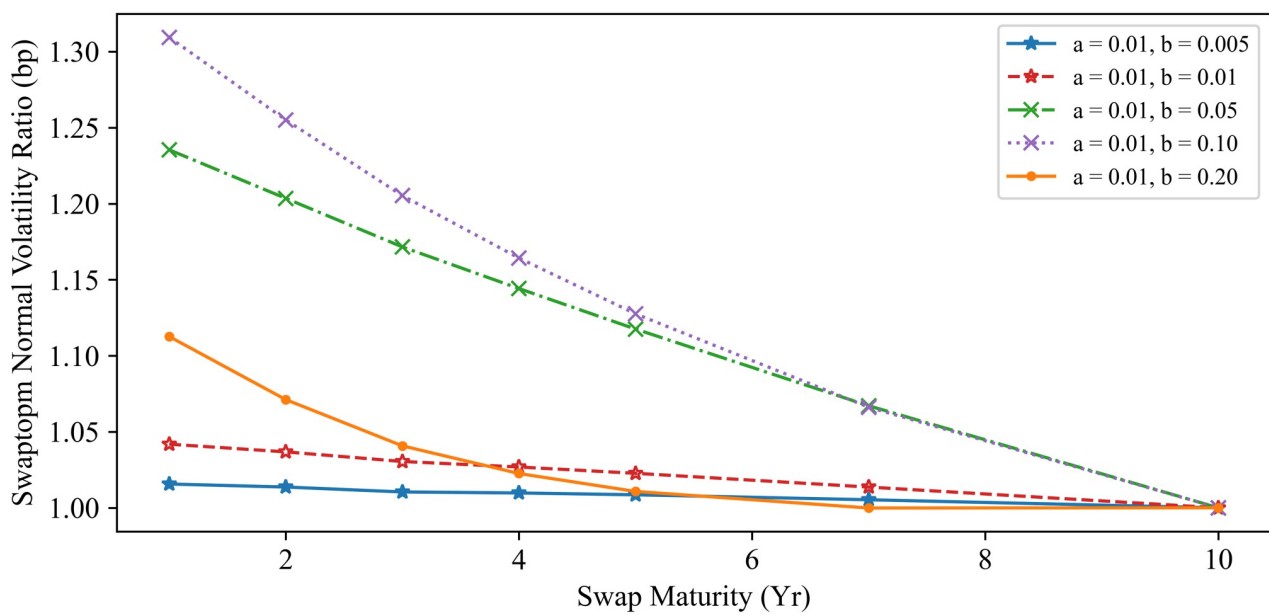

## Panel B. Changing σ, η at fixed a, b

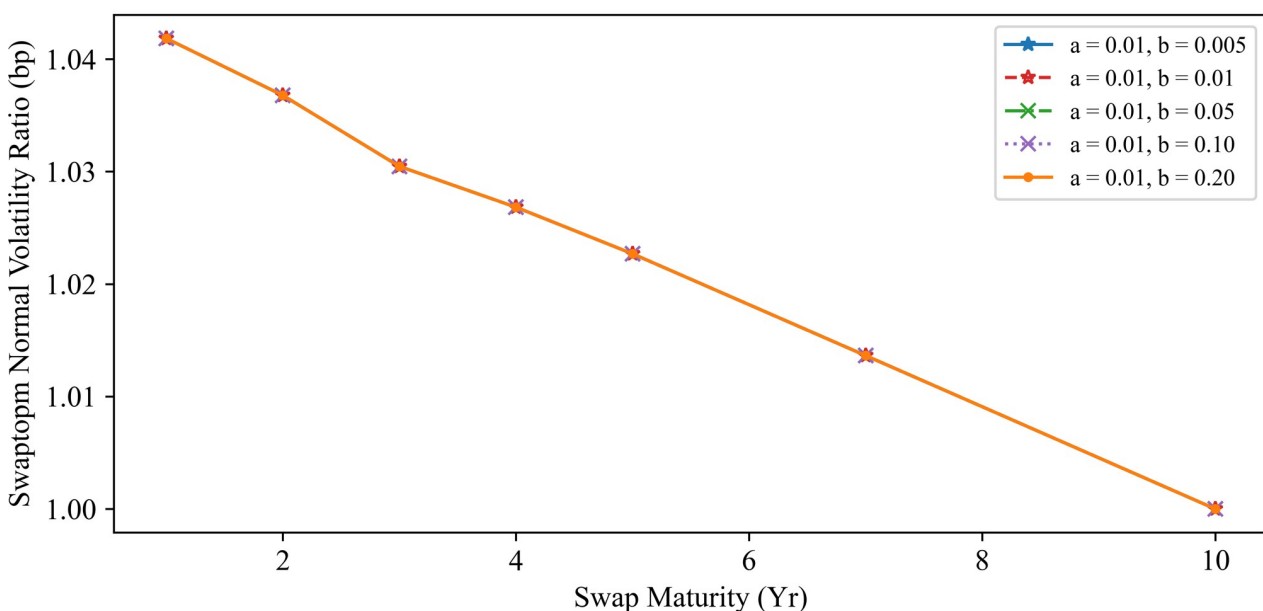

**Fig 4. Swaption normal volatility ratios to swap maturity of 10Y with option expiry of 10Y.** Panel A shows the swaption normal volatility ratios to swap maturity of 10Y and other tenors under various mean reversions and fixed volatility parameters. All swaptions have an option expiry of 10Y. The swaption normal volatility ratio curve moves downward as the mean reversion parameters increase. Panel B shows the swaption normal volatility ratios to swap maturity of 10Y and other tenors under various volatility parameters when the mean reversions and the ratio of volatility parameters are fixed. The swaption normal volatility ratio curves for the various volatility parameters are above the same line. That is, the swaption normal volatility ratios with equal swap maturity depend entirely on the mean reversions.

In summary, the volatility ratios of swaptions with the same option expiry depend mainly on the mean reversion parameters if $\mathcal{C} = \sigma/\eta$ is fixed. This observation inspires the second calibration strategy where the mean reversions and volatility parameters are calibrated separately.

**Table 2. Calibration basket of "20NC1Yr callable CMS spread floater".** This table presents a calibration basket comprising 36 swaptions. The basket contains co-terminal swaption and comprises two indexes that make up the CMS spread, a swaption with a swap maturity of 20Y and 10Y. The option expiry of the basket can be either the early exercise date of the callable product or option expiry quoted in the swaption market. In this study, we use the option expiry quoted in the swaption matrix.

| | Swap Maturity | | |
|---|---|---|---|
| Option expiry | Co-terminal Swaption | SpreadIndex1 | SpreadIndex2 |
| 1M | 19Y11M | 10Y | 2Y |
| 3M | 19Y9M | 10Y | 2Y |
| 6M | 19Y6M | 10Y | 2Y |
| 9M | 19Y3M | 10Y | 2Y |
| 1Y | 19Y | 10Y | 2Y |
| 18M | 18Y6M | 10Y | 2Y |
| 2Y | 18Y | 10Y | 2Y |
| 3Y | 17Y | 10Y | 2Y |
| 4Y | 16Y | 10Y | 2Y |
| 5Y | 15Y | 10Y | 2Y |
| 7Y | 13Y | 10Y | 2Y |
| 10Y | - | 10Y | 2Y |
| 19Y (last call date) | 1Y | - | - |

## Choice of instruments

Let us consider the case of calibrating KRW swaptions. Our data comprised 84 swaptions. This does not refer to 84 degrees of freedom. Instead, the user can select a subset of swaptions to estimate the parameters of the model. Future tests will consider local calibration, a popular strategy among practitioners. This calibration selects swaptions with a fixed co-terminal

**Table 3. Calibration basket for step (1) of the two-step approach.** This table presents the calibration basket for step (1) of the two-step approach. Based on the volatility with swap maturity of 10Y, the longest swap maturity in the market quote, the basket comprises the swaption volatility ratio with swap maturity of 1Y, and the shortest swap maturity on the market. In our test case, the non-call period is 1-year; thus, we use an option expiry from 1Y to 10Y. Moreover, we include the ratio of volatility with swap maturity of 5Y in the basket.

| Option expiry | Swap Maturity in denominator | Swap Maturity in numerator | Swaption Normal Volatility Ratio |
|---|---|---|---|
| 1Y | 10Y | 1Y | $\mathcal{V}^{mkt}(1Y,10Y)/\mathcal{V}^{mkt}(1Y,1Y)$ |
| | | 5Y | $\mathcal{V}^{mkt}(1Y,10Y)/\mathcal{V}^{mkt}(1Y,5Y)$ |
| 2Y | 10Y | 1Y | $\mathcal{V}^{mkt}(2Y,10Y)/\mathcal{V}^{mkt}(2Y,1Y)$ |
| | | 5Y | $\mathcal{V}^{mkt}(3Y,10Y)/\mathcal{V}^{mkt}(3Y,5Y)$ |
| 3Y | 10Y | 1Y | $\mathcal{V}^{mkt}(3Y,10Y)/\mathcal{V}^{mkt}(3Y,1Y)$ |
| | | 5Y | $\mathcal{V}^{mkt}(3Y,10Y)/\mathcal{V}^{mkt}(3Y,5Y)$ |
| 4Y | 10Y | 1Y | $\mathcal{V}^{mkt}(4Y,10Y)/\mathcal{V}^{mkt}(4Y,1Y)$ |
| | | 5Y | $\mathcal{V}^{mkt}(4Y,10Y)/\mathcal{V}^{mkt}(4Y,5Y)$ |
| 5Y | 10Y | 1Y | $\mathcal{V}^{mkt}(5Y,10Y)/\mathcal{V}^{mkt}(5Y,1Y)$ |
| | | 5Y | $\mathcal{V}^{mkt}(5Y,10Y)/\mathcal{V}^{mkt}(5Y,5Y)$ |
| 7Y | 10Y | 1Y | $\mathcal{V}^{mkt}(7Y,10Y)/\mathcal{V}^{mkt}(7Y,1Y)$ |
| | | 5Y | $\mathcal{V}^{mkt}(7Y,10Y)/\mathcal{V}^{mkt}(7Y,5Y)$ |
| 10Y | 10Y | 1Y | $\mathcal{V}^{mkt}(10Y,10Y)/\mathcal{V}^{mkt}(10Y,1Y)$ |
| | | 5Y | $\mathcal{V}^{mkt}(10Y,10Y)/\mathcal{V}^{mkt}(10Y,5Y)$ |

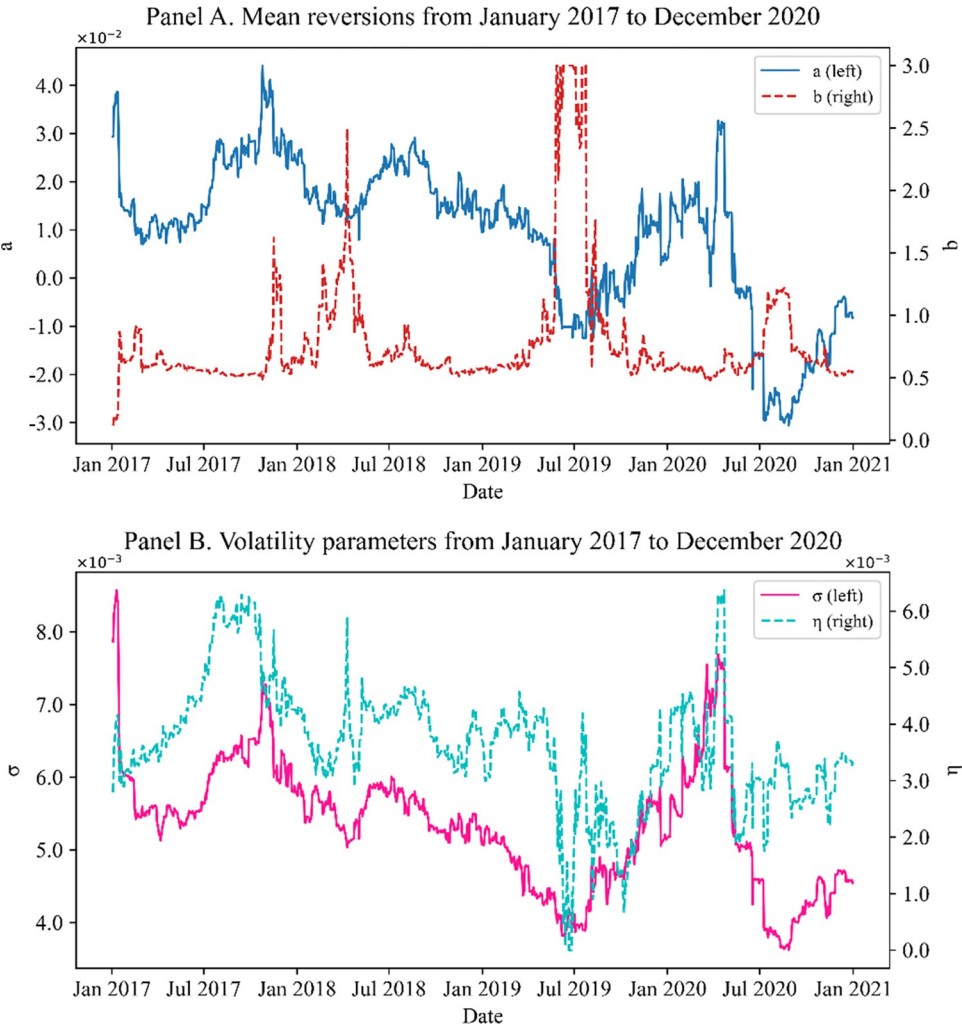

**Fig 5. (Method I) model parameters calibrated to 20Y co-terminal and 10Y-2Y spread with ρ = −0.9.** Panel A shows the mean reversions calibrated with step (1) of the two-step approach, and panel B shows the volatility parameters calibrated with step (2). A significant change occur in the 2nd and 3rd quarters of 2019 in all panels. This is because the swap rates declined owing to cuts in the US and Korea's benchmark rate, and swaption normal volatility rose.

(option expiry + swap maturity) and an underlying swap. We consider the 20Y co-terminal swaption and the 10Y-2Y spread index. Table 2 summarizes the results of this study.

For Method II in Section 3.4, we need an additional calibration basket for the mean reversions. The instruments for the calibration basket comprise two swaptions with the same option expiry and different swap tenors. As shown in Table 3, we choose 1-, 2-, 3-, 4-, 5-, 7-, 10-years for option expiry, 10-years for long swap maturity, and 1-year for short swap maturity. Our test uses four-year data from January 2017 to December 2020, where we perform the calibration daily.

## Co-terminal 20Y and 10Y-2Y spread index

In the simplest case of constant $a$, $b$, $\sigma$, and $\eta$, we can find a typical behavior of overfitting on the mean-reversion side. This section compares the results of the three aforementioned methods.

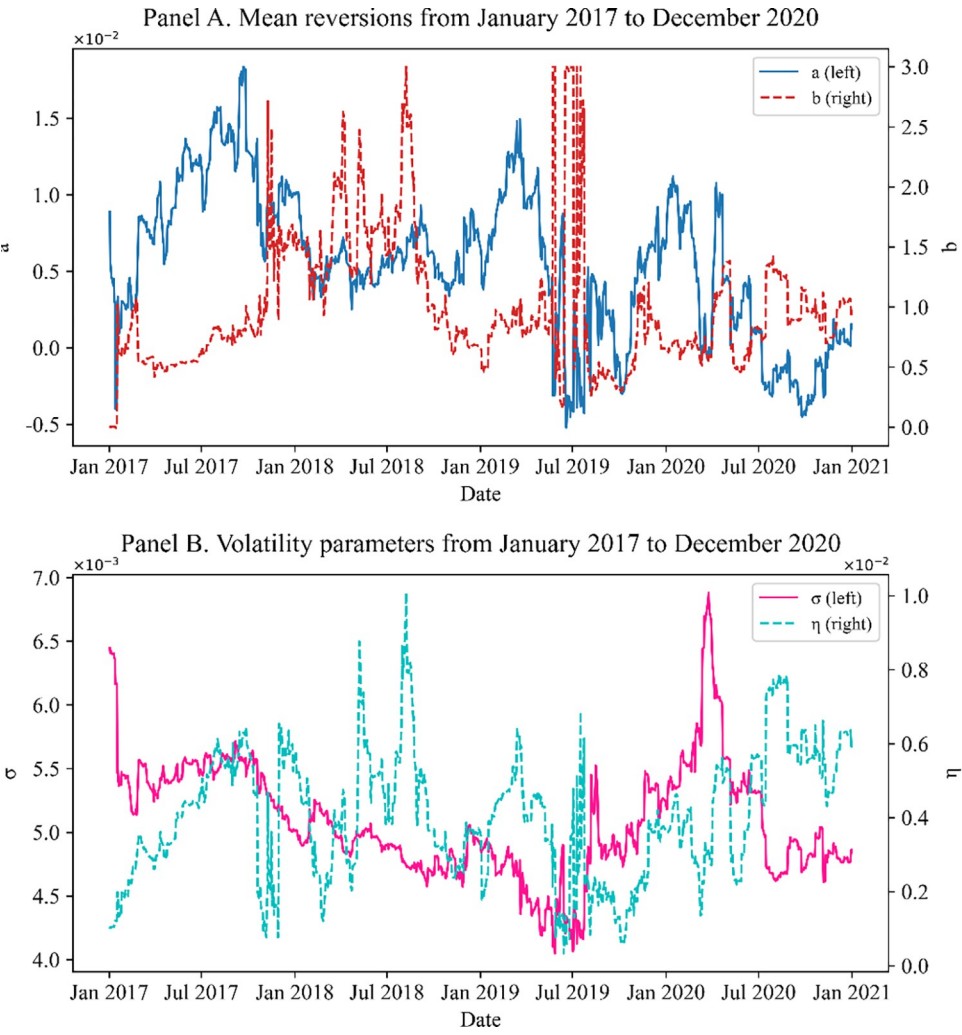

**Fig 6. (Method II) model parameters calibrated to 20Y co-terminal and 10Y-2Y spread with ρ = −0.9.** Panels A and B show the mean reversions and volatility parameters, respectively, calibrated with the one-step method. The mean reversions and volatility parameters change significantly every day. Many significant changes occur (e.g., in the 2nd and 3rd quarters of 2019).

Figs 5–7 show the parameters obtained by Methods I, II, and III, respectively. In all cases except Method III, significant changes in mean reversions are observed in the 2nd and 3rd quarters of 2019. The swap rates decline owing to cuts in the benchmark rates of Korea and the US, and the swaption normal volatility rose.

In Fig 5, the mean reversion $a$ and the volatility parameter $\sigma$ behave similarly. However, we can see that the other volatility parameter $\eta$ changes significantly when the mean reversion $b$ has a significant change. The parameters reflect changes in the market when market events such as falling benchmark interest rates occur.

Figs 6 and 7 show that the results from Methods II and III can change significantly depending on market conditions, even without large market events. Naturally, the more the parameters to be calibrated, the more the sensitivity of the results to even small market changes (owing to the curse of dimensionality); numerous local minima exist in the optimization algorithm. Therefore, the parameters significantly depend on the initial seeds. This problem affects the measure of sensitivity, such as the delta and vega. For structured products, no closed form

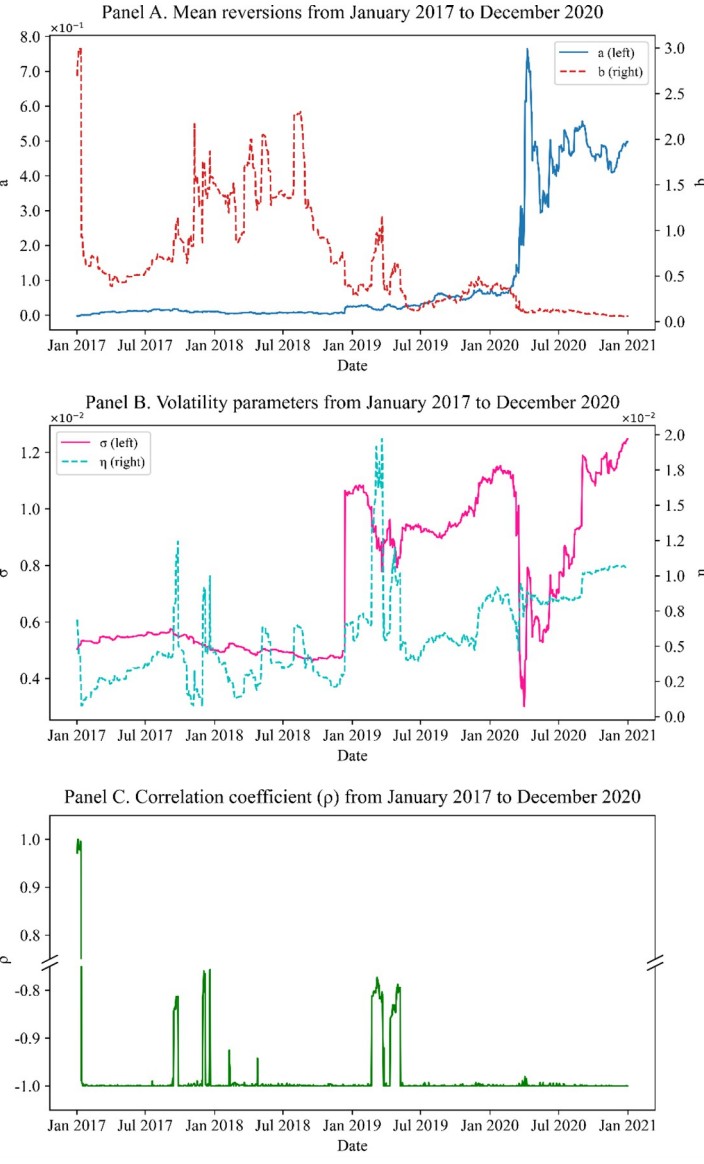

**Fig 7. (Method III) model parameters (*ρ* included) calibrated to 20Y co-terminal and 10Y-2Y spread.** Panels A, B, and C show the mean reversions, volatility parameters, and correlation coefficient, respectively, calibrated with the one-step method. Here, the correlation coefficient stands out and is sensitive to changes in market data. Whereas, the mean reversion parameter hardly moves, the others change dramatically.

exists for the sensitivity. Therefore, we typically change the market data to numerically compute the sensitivity. Here, if the model parameters are overly sensitive when the interest rate increases by 1bp, delta or vega are not stable, which results in poor risk management. Generally, we conclude that the two-step method (Method I) produces more stable results than the alternatives, proving the practical validity of our suggestion.

## Conclusion

This paper suggests an efficient approximation of the swaption normal volatility in the G2PP.

We analyze the one-step approach and the novel two-step approach to estimate the model parameters using the approximation. The one-step method calibrates all parameters

simultaneously. The two-step approach estimates the mean reversion in the separation of the other volatility parameters of the G2PP. We compare the two-step method with the one-step method in the interest rate market of Korea and the US. We find that the parameter estimates from our two-step approach are more stable than those from the one-step method.

This study adds two contributions to the literature. First, we provide detailed documentation and numerical examples of calibration methods for the G2PP model that the existing literature has disregarded. Our findings will be a useful starting point for practitioners to implement their calibration because it would save trial and error attempts. Second, we dispute some negative opinions about the G2PP model [10]. For example, it is widely known that the G2PP model cannot fit the swaption matrix well and becomes unstable as the number of parameters increases. We find that the two-step approach eliminates these problems.

This study has three limitations. First, we do not aim to identify the best calibration method. Instead, we compare the three calibration methods using constant parameters. Users can select a calibration method depending on their purpose and preference for fitting quality, run time, and simplicity of implementation. Nevertheless, a low degree of freedom in the optimization provides more stable results, and we show how to calculate mean reversions with a lower degree of freedom. We use the Levenberg–Marquet algorithm for the optimization method. Whereas the results from the optimization algorithm are usually sensitive to seed values owing to the nature of the algorithms, we can overcome such problems with the proposed two-step approach, which has a smaller dimension.

Second, although we propose and analyze a constant parameter approach, other creative alternatives are possible. Future studies can investigate other parameter settings with different degree of freedom. For example, one can modify the two-step approach, assuming constant mean reversions in the first step and piecewise constant volatility parameters in the second step.

Third, we use calibration to calculate the sensitivity and the price of the derivatives. To evaluate a single product, one would need as many as hundreds of calibration processes. In the future, deep learning techniques can be applied to this calibration process, whereas this study highlights only the two-step approach.

## Supporting information

**S1 Appendix. Model description.**
(DOCX)

**S2 Appendix. Other numerical results (USD currency).**
(DOCX)

## Acknowledgments

Do not include funding or competing interests information in Acknowledgments.

## Author Contributions

**Conceptualization:** Myeongsu Choi.

**Methodology:** Myeongsu Choi.

**Supervision:** Hyoung-Goo Kang.

**Visualization:** Hyoung-Goo Kang.

**Writing – original draft:** Myeongsu Choi.

**Writing – review & editing:** Hyoung-Goo Kang.

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
